# Kinetic Analysis of the Curing Process of Biobased Epoxy Resin from Epoxidized Linseed Oil by Dynamic Differential Scanning Calorimetry

**DOI:** 10.3390/polym13081279

**Published:** 2021-04-14

**Authors:** Diego Lascano, Alejandro Lerma-Canto, Vicent Fombuena, Rafael Balart, Nestor Montanes, Luis Quiles-Carrillo

**Affiliations:** 1Technological Institute of Materials (ITM), Universitat Politècnica de València (UPV), Plaza Ferrándiz y Carbonell 1, 03801 Alcoy, Spain; dielas@epsa.upv.es (D.L.); allercan@epsa.upv.es (A.L.-C.); rbalart@mcm.upv.es (R.B.); nesmonmu@upvnet.upv.es (N.M.); luiquic1@epsa.upv.es (L.Q.-C.); 2Escuela Politécnica Nacional, Quito 17-01-2759, Ecuador

**Keywords:** epoxidized linseed oil, kinetic analysis, biobased epoxy resin, flax

## Abstract

The curing process of epoxy resin based on epoxidized linseed oil (ELO) is studied using dynamic differential scanning calorimetry (DSC) in order to determine the kinetic triplet (*E_a_*, f(α) and *A*) at different heating rates. The apparent activation energy, *E_a_*, has been calculated by several differential and integral isoconversional methods, namely Kissinger, Friedman, Flynn–Wall–Ozawa (FWO), Kissinger–Akahira–Sunose (KAS) and Starink. All methods provide similar values of *E_a_* (between 66 and 69 kJ/mol), and this shows independence versus the heating rate used. The epoxy resins crosslinking is characterized by a multi-step process. However, for the sake of the simplicity and to facilitate the understanding of the influence of the oxirane location on the curing kinetic, this can be assimilated to a single-step process. The reaction model has a high proportion of autocatalytic process, fulfilling that *α_M_* is between 0 and *α_p_* and *α_M_* < $\alpha_{p}^{\infty }$. Using as reference the model proposed by Šesták–Berggren, by obtaining two parameters (*n* and *m*) it is possible to obtain, on the one hand, the kinetic parameters and, on the other hand, a graphical comparison of the degree of conversion, *α*, versus temperature (*T*) at different heating rates with the average *n* and *m* values of this model. The good accuracy of the proposed model with regard to the actual values obtained by DSC gives consistency to the obtained parameters, thus suggesting the crosslinking of the ELO-based epoxy has apparent activation energies similar to other petroleum-derived epoxy resins.

## 1. Introduction

Flax is one of the industrial crops that, thanks to its versatility, has been used since ancient times both in food and in clothes manufacturing for approximately 30,000 years [1]. Nowadays, flax derivatives are continuously gaining importance, due to the productive and economic nature of its sowing. For this reason, the largest producers of flax (Russia, China, Kazakhstan, Argentina and Canada) have increased their production capability [2,3]. In addition to the main uses related to flax crops, namely textile fibers and seeds for food industry, while new uses are being developed for other by-products or co-products linked to the flax industry. In the last decade, flax oil, biodiesel and flax flour have found increasing applications in the market [4,5,6].

One of the most interesting by-product is flax oil (or linseed oil), which represents approximately 45% of the seeds [3,7]. The main fatty acids are linolenic acid (56.6%), oleic acid (19.1%) and linoleic acid (15.3%), with three, one and two unsaturations, respectively [2]. In addition to foods and feeding applications, linseed oil offers very attractive industrial applications due to its high unsaturated fatty acid content. Among other uses, it has been reported the use of linseed oil and derivatives into paints, surface protection, cosmetic products and biodiesel generation [8,9,10].

By taking into account the high degree of unsaturated carbons contained in the main fatty acids, which represents about 73%, its chemical modifications opens a broad potential for industrial applications [11,12]. Some of the simplest and cost-effective modification processes are epoxidation and maleinization. Fombuena et al. [13], reported the development of high renewable content green composites, with more than 78 wt.% of flax-derived materials, namely biobased epoxidized linseed oil (ELO) as epoxy resin, maleinized linseed oil (MLO) as hardener and flax fibers as reinforcement. Liminana et al. and Carbonell–Verdu et al. [14,15] reported the potential of MLO as compatibilizer between a natural reinforcement (almond shell flour) and a thermoplastic polyester such as poly (butylene succinate) (PBS); they showed improved polymer-particle interaction due to presence of MLO and somewhat plasticization effect provided by this maleinized-derivative from linseed oil.

As above-mentioned, one of the most common chemical modification on unsaturated fatty acids is epoxidation. This increases the reactivity of the linseed oil since the oxirane rings are much more reactive than unmodified unsaturations in fatty acids [16,17]. Epoxidized linseed oil is commercially available and finds industrial applications as secondary plasticizer in poly (vinyl chloride) plastisols and partially biobased epoxy resins [18,19]. ELO-based epoxies can represent a sustainable solution to traditional epoxy resins, generally derived from petroleum, which has a large impact on increasing the carbon footprint, and can also be toxic to humans [20]. The epoxidation process takes advantage of the unsaturations contained in the fatty acids of linseed oil by converting them into oxirane rings [21]. Miyagawa et al. [22] formulated a resin based on bisphenol F with the presence of ELO and an anhydride as hardener. The results showed lower thermomechanical and thermal properties at higher ELO content. Boquillon and Fringant [23] formulated anhydride cured ELO-based resins catalyzed with amines and imidazoles. In this case, the presence of imidazole catalyst improved the obtained results since it improved the crosslinking process with the subsequent improvement on mechanical performance.

This crosslinking process between an epoxy resin and the hardener provides an exothermic reaction, with an important release of heat that can contribute to increase the temperature, and subsequently, the possibility to provide an autocatalytic process. In previous works, the curing kinetics of a bisphenol A-based resin with a content of 31% by weight of ELO was studied, and it was observed that the mechanical properties obtained were similar to those of a conventional petroleum-derived epoxy resin [24]. In the present work, the curing kinetics of an ELO-based epoxy resin crosslinked with a liquid anhydride is studied by differential scanning calorimetry (DSC) using non-isothermal runs. By means of differential and integral isoconversional models, the apparent activation energy, *E_a_*, the reaction model, f(α) and the pre-exponential factor, *A*, are determined. The main aim of this work is to compare the kinetic triplet of the ELO-based epoxy resin with other petroleum-derived epoxies to assess the effect of non-terminal oxirane groups in ELO with regard to conventional epoxy resins with epoxy groups located in terminal and more readily available positions. Furthermore, through the comparison of the experimental functions *y*(*α*) and *z*(*α*), with standard master plots and the determination of the *m* and *n* parameters of Šesták–Berggren model (*SB*(*m*,*n*)), the influence of autocatalysis process and the accuracy of the kinetic triplet parameters are determined.

## 2. Experimental

### 2.1. Materials

Commercial epoxidized linseed oil was used as base epoxy resin to carry out a kinetic study. ELO was obtained from Traquisa S.L. (Barberá del Vallés, Barcelona, Spain). This biobased epoxy stands out for having a fatty acid composition of: Stearic acid (3–5%), palmitic acid (5–7%), linoleic acid (14–20%), oleic acid (18–26%) and linolenic acid (51–56%). It also presents an average molecular weight of 1.038 g/mol triglyceride, and a density of 1.05–1.06 g/cm^3^ at 20 °C. The epoxy equivalent weight (EEW) of the ELO-based epoxy resin was 178 g/equiv. A liquid at room temperature cyclic anhydride, namely, methyl nadic anhydride (MNA), with an anhydride equivalent weight (AEW) of 178.18 g/equiv was employed. This anhydride-based hardener was employed due to epoxy/anhydride systems have higher glass transition and less shrinkage, interesting properties if the objective is the use of ELO resin in industrial applications [25]. The epoxy to anhydride ratio (EEW:AEW) was set to 1:1. The crosslinking system also consisted of 2 wt % of 1-methyl imidazole which acts as accelerator, and 0.8 wt.% glycerol that plays a key role as a catalyst/initiator. All these products were supplied by Sigma Aldrich (Sigma Aldrich, Madrid, Spain). To carry out the kinetic study, all the components of the crosslinking system were weighed on a balance with an accuracy of 0.01 grams, and were mixed at room temperature and stirred vigorously until obtaining a homogeneous mixture [26]. The liquid nature of the MNA anhydride allows good mixing at room temperature thus avoiding precipitation and the starting of the crosslinking process.

### 2.2. Kinetic Study by Differential Scanning Calorimetry (DSC)

The kinetic analysis of the curing process was carried out by following the crosslinking enthalpy which allowed calculating the degree of advance (*β*). The evolution of the crosslinking enthalpy was obtained in a DSC Mettler–Toledo DSC 821e (Metler–Toledo S.A.E., Barcelona, Spain). After the corresponding mixing of all components and homogenization, a small drop of the liquid mixture containing between 10–12 mg was placed into a standard aluminum crucible (40 µL) following the recommendations of Vyazovkin et al. [27]. Samples were sealed with a press and tested using a non-isothermal heating program from 20 °C up to 300 °C at different heating rates (10, 20, 30, 40 °C/min) and N_2_ atmosphere with constant flow rate of 30 mL/min. Parameters such as the degree of curing, and the maximum curing rate (*T_p_*) can be easily obtained applying this methodology [28].

### 2.3. Theoretical Background

Kinetic studies are based on the principle that the degree of conversion, *α*, can be obtained (using a non-isothermal heating) by the heat produced by the curing process with respect to time, divided by the total heat of curing of the resin as presented in Equation (1) [29]:(1)$$\alpha =\frac{\Delta H_{t}}{\Delta H_{T}}$$

Taking this into account, the conversion rate $\frac{d\alpha }{dt}$ can be expressed according to Equation (2):(2)$$\frac{d\alpha }{dt}=k\left(T\right)\cdot f\left(\alpha \right)$$
where *k*(*T*) is the rate coefficient, depending on the reaction temperature and f(α) represents the reaction model. The Arrhenius equation (Equation (3)) expresses the dependence of the temperature:(3)$$k\left(T\right)=A\cdot e^{\frac{-E_{a}}{R\cdot T}}$$
where *E_a_* stands for the apparent activation energy, *A* represents the pre-exponential factor, *R* is the universal gas constant (8.314 J mol^−1^ K^−1^) and *T* is the absolute temperature. Combining Equations (2) and (3), the general expression in the kinetic analysis of the curing resin is obtained Equation (4):(4)$$\frac{d\alpha }{dt}=A\cdot e^{\frac{-E_{a}}{R\cdot T}}\cdot f\left(\alpha \right)$$

The parameters *E_a_*, *f*(*α*) and *A* are known as the kinetic triplet. The determination of this triplet is of huge importance to understand the kinetics of crosslinking process. The apparent activation energy (*E_a_*) can be obtained through different methods, the accuracy of which depends on the fulfilment of approaches such as the type of transformation process being carried out, single or multi-step process. Due to its simplicity, Kissinger method is one of the most used [30]. The apparent activation energy (*E_a_*) is evaluated based on the peak temperature (*T_p_*) measurements at maximum curing rate, taking into account that Kissinger method is only valid for a one-step process [31]. The Kissinger method follows Equation (5):(5)$$\ln\left(\frac{\beta }{T_{p}^{2}}\right)=\ln\left(\frac{A\cdot R}{E_{a}}\right)-\frac{E_{a}}{R\cdot T_{p}}$$

By plotting $\ln\left(\frac{\beta }{T_{p}^{2}}\right)$ versus $\frac{1}{T_{p}}$, the apparent activation energy is obtained through the slope of the linear fit. As a main drawback, the Kissinger method only provides a single value of the *E_a_* at the peak temperature and cannot follow the evolution of *E_a_* with the conversion α. Despite this, several authors have tried to overcome this drawback, such as Farjas et al. [32]. This method complements the maximum peak values with additional values such as the time corresponding to the full width at half maximum (∆t_FWHM_), which is more sensitive to the existence of multiple processes.

An alternative to the Kissinger method are the isoconversional methods, which, together with their simplicity, provide an accurate estimation of the *E_a_*. Vyazovkin et al. proposes different isoconversional methods, evaluating the *E_a_* throughout the conversion process with the advantage of not having to assume a reaction model [33]. Isoconversional methods can be differential or integral. The expressions of these methods, regardless of their type, can be obtained from Equation (4), applying the principle of isoconversion, results in Equation (6):(6)$${\left[\frac{\partial \ln\left(\frac{d\alpha }{dt}\right)}{\partial \frac{1}{T}}\right]}_{\alpha_{i}}=-\frac{E_{a,\;\alpha i}}{R}$$

As above-mentioned, the adequate estimation of apparent activation energy (*E_a_*) determines the accuracy of the kinetic study of the crosslinking process and here is where isoconversional integral methods play an important role. Based on integration of Equation (4), the resulting expression is observed in Equation (7):(7)$$\overset{\alpha }{\underset{0}{\int }}\frac{d\alpha }{f\left(\alpha \right)}=A\cdot \overset{t}{\underset{0}{\int }}e^{\frac{-E_{a}}{R\cdot T}}dt$$

As an example of differential isoconversional method, the Friedman method [34], takes natural logarithms. Rearranging terms for Equation (8), this method assumes that the reaction model stays constant throughout the full conversion process. The advantage of this method is that does not make approaches but provides valuable information of the actual crosslinking process [35].
(8)$$\ln\left(\beta \cdot \frac{d\alpha }{dT}\right)=\ln\left[A\cdot f\left(\alpha \right)\right]-\frac{E_{a}}{R\cdot T_{\alpha }}$$

Different isoconversional integral methods with different accuracies on the *E_a_* estimation have been developed by some authors. The accuracy of these methods depends on the numerical approximation obtained by solving the temperature-dependent integral term of the Arrhenius constant (temperature integral). Among others, the usefulness of the Flynn–Wall–Ozawa (FWO) Equation (9) [36,37], Kissinger–Akahira–Sunose (KAS) Equation (10) [38] or Starink methods Equation (11) [39] should be noted. This last method (Starink method), playing with the optimization of the mathematical parameters obtain a slope of $-1.008\cdot \frac{E_{a}}{R\cdot T_{\alpha }}$ when $\ln\left(\frac{\beta }{T_{\alpha }^{1.92}}\right)$ is plotted versus $\frac{1}{T_{\alpha }}$, while FWO and KAS methods provide a slope of −1.000 and −1.052, respectively. The mathematical approximation done by Starink method has demonstrated higher accuracy than FWO and KAS methods.
(9)$$\ln\left(\beta \right)=C-1.052\cdot \frac{E_{a}}{R\cdot T_{\alpha }}$$
(10)$$\ln\left(\frac{\beta }{T_{\alpha }^{2}}\right)=C-1.000\cdot \frac{E_{a}}{R\cdot T_{\alpha }}$$
(11)$$\ln\left(\frac{\beta }{T_{\alpha }^{1.92}}\right)=C-1.008\cdot \frac{E_{a}}{R\cdot T_{\alpha }}$$

On the other hand, Vyazovkin et al. [33], with the aim of reducing, as far as possible, the deviation of *E_a_*, recommends that the values of the conversion degree should be taken from α = 0.05 to α = 0.95 by increasing steps of 0.05. In this way, expanding the analysis range, the accuracy of the estimation of *E_a_* is increased. Malek [40], proposes a method by determining the reaction model calculating two functions *y*(*α*) and *z*(*α*,) defined as shown in Equations (12) and (13):(12)$$y\left(\alpha \right)=\frac{d\alpha }{dt}\cdot e^{x}=A\cdot f\left(\alpha \right)$$
(13)$$z\left(\alpha \right)=\pi \left(x\right)\cdot \left(\frac{d\alpha }{dt}\right)\cdot \frac{T}{\beta }$$

Looking for similarities between the FWO method Equation (9) and the method proposed by Malek, the term *x* is equivalent to $\frac{E_{a}}{R\cdot T_{\alpha }}$, and *A* to f(α), that represents the function reaction model, during the crosslinking process. Regarding the z(α), it is determined by the expression of the temperature integral π(x), which can be obtained using the 4th rational expression proposed by Senum and Yang [41]. On the other hand, Flynn [42], suggested a correction in the expression as it is shown in Equation (14), that has been obtained from experimental data:(14)$$\pi \left(x\right)=\frac{x^{3}+18x^{2}+86x+96}{x^{4}+20x^{3}+120x^{2}+240x+120}$$

Table 1 summarizes several kinetic models with expressions of their characteristic function *f*(*α*), whose representation of the selected reaction model determines the accuracy of the linear fit to obtain *E_a_*.

The success of a correct kinetic model that accurately predicts the complex mechanism of curing of vegetal oil resins derives from several factors, among the main ones the use of appropriate methods for the determination of the kinetic triplet as well as the correct data collection. In the current literature, several articles related to the kinetic study of different vegetable oils can be found. Due to the similarity in the number and position of epoxide groups, the studies carried out on soybean oil are noteworthy [43,44,45,46]. Regarding epoxidized linseed oil, studies with the use of photocrosslinkers have been carried out for the use of ELO as a coating for wood [47]. Kinetic studies have been done which carried out by means of crosslinking with acid hardeners, supporting the use of isoconversional methods for the determination of the kinetic parameters [48]. In this study, catalysts and accelerators are not employed restricting the extrapolation of the results to industrial scale, where these elements are key in process acceleration and scale-up. In another example, Lascano et al. in a previous study, determined the accuracy of different isoconversional methods by comparing different methods, but in this case, applied to a resin based on diglycidyl ether of bisphenol A (DGEBA) cross-linked with amine hardener [24]. On the other hand, in a study by Mahendran et al. different isoconversional methods such as Friedman (FR), KAS and Vyazovkin (VA) were studied on an ELO resin cross-linked with ahnidrides [49], but in contrast to the present article, the study of the possible autocatalytic contribution in the cross-linking process, using the Šesták–Berggren methodology, has not been carried out. In view of the above-mentioned aspects, the present article performs a thorough comparison of the different isoconversional methods in order to determine their adequacy for an ELO-based epoxy resin crosslinked with anhydrides. In addition, the possible autocatalytic contribution in the crosslinking process is studied, as well as the possible influence of the location of the different epoxy groups in the determination of these parameters.

## 3. Results and Discussion

Dynamic differential scanning calorimetry was employed to analyze the curing kinetics of the epoxy resin based on epoxidized linseed oil. Samples were subjected to a single step heating cycle, obtaining the total enthalpy released (Δ*H_T_*) during the crosslinking process. Figure 1 shows the DSC plot representation of the curing process of epoxy resin based on epoxidized linseed oil at different heating rates (10, 20, 30 and 40 °C/min). The temperature for the maximum curing rate or peak temperature (*T_p_*), was determined analyzing the peak during the heating cycle.

DSC results are summarized in Table 2. A clear increasing tendency can be observed for the temperature peak for the maximum curing rate (*T_p_*). This changes from 192.8 °C using a heating rate (*β*) of 10 °C/min, and exceeds 223 °C for the highest heating rate in this study (40 °C/min). The acceleration of the crosslinking reaction, increasing *β*, gives rise to a three-dimensional crosslinked net structure which is responsible for reducing chain mobility and, subsequently, hindering the free diffusion, this having an effect on both temperature for the maximum rate (*T_p_*) and the total released enthalpy (Δ*H_T_*), as some authors as Carbonell et al. and Rocks et al. suggested [50,51]. On the other hand, the glass transition temperature (*T_g_*) can vary depending on the heating rate and the post-curing process. Using a dynamic-mechanical thermal analysis (DMTA), Samper et al. demonstrated a glass transition temperature (*T_g_*) of the cured ELO-MNA system without post-curing process close to 127 °C, with a heating rate of 5 °C/min [52]. Therefore, knowing that higher heating rates can cause an increase in *T_g_*, it is expected that these values are higher than the 127 °C obtained in the study by Samper et al., and lower than the 148 °C obtained when a post-cure of 2 h at 160 °C is applied.

Figure 2a shows the plot representation of conversion (*α*) versus absolute temperature (*T*). It is important to remark that as the heating rate increased, the sigmoidal curves are displaced to higher temperature. As the geometric shape and the distance between curves are almost the same, it is possible to suggest that the heating rate (*β*) does not affect the reaction model *f*(*α*) even though the crosslinking process of an epoxy resin is not a simple process. Figure 2b shows the plot representation of the curing rate ($\frac{d\alpha }{dT}$) as a function of the conversion (*α*). Once again, it can be seen how increasing the heating rate (*β*) from 10 to 40 °C/min, the shape of the curve remains almost invariable, thus suggesting the same reaction model *f*(*α*) for the entire range studied. It is worthy to note that the curing or crosslinking process of an epoxy resin is not a typical single-step reaction. It consists of several overlapped processes related with the studies initiated by authors such as Fischer and Tanaka and Kakiuchi in the 1960s [53,54,55]. Multiple reactions occur at the same time, related with the presence of a proton donor, which can start the reaction with the epoxy groups, and the anhydride-based hardener [56]. Nevertheless, the overall process, as shown in Figure 1, can be assimilated to a single-step process for the sake of simplicity and comparison with kinetic parameters of other petroleum-derived epoxy resins and assess the effect of the oxirane location on the curing kinetics.

Within the goal to determinate the kinetic triplet (*E_a_*, f(α) and *A*), the apparent activation energy is one of the most important parameters. Kissinger method is one of the simplest methods Equation (5), since it can be used independently of the transformation process, in this case, a curing process. Despite this, this method assumes a single-step process and, as above-mentioned, the crosslinking process of an epoxy resin is a multi-step process, as such the Kissinger method gives an initial approximation of the kinetic parameters. As above-mentioned, the accuracy of these methods depends on the type of reaction that is taking place, being the accuracy enhanced if the reaction occurs in a single step for the different heating rates (*β*). For this reason, the conversion at the maximum reaction rate $\alpha_{p}$, must be very similar, independently of the heating rate (*β*) used. In the previous Figure 2b $\alpha_{p}$ is shown in the highest point of the plot representation. These values are giving as average value of $\alpha_{p}$, 0.5733 ± 0.02 with values very similar to the different heating rates used and values reported by literature [24,57].

Using the Kissinger method, Figure 3a represents the linear fitting, whose slope correspond to the *E_a_*. In this case the value obtained is 70.07 ± 0.066 kJ/mol, which is a typical value consistent with the literature data of epoxy resin (*E_a_* = 70 kJ/mol) [57]. As above-mentioned, one of the main drawbacks of the Kissinger method is the analysis of one-step reaction model using only one point equivalent to maximum reaction temperature [58]. For this reason, following the approach suggested by Farjas et al. the variation of ln Δt_FWHM_ versus *1*/(*T_p_*) has been plotted in Figure 3b. In the case to obtain values of *E_a_* very similar to Kissinger method it would indicate that the reaction is indeed carried out in one-step. The slope obtained after linear fitting is 69.12 ± 0.039 kJ/mol being practically the same, which corroborates the correct approximation done with the use of Kissinger method.

Notwithstanding the ease of application of the Kissinger method, it does not fail to give a single value for *E_a_*. Thus, to evaluate possible change of *E_a_* with the conversion, the use of isoconversional methods (both differential and integral methods) is needed. Results obtained by Friedman, FWO, KAS and Starink methods are gathered in Table 3. In The first approach done by Friedman method (differential method), the apparent energy is determined from the slope of the plot $\ln\left(\frac{d\alpha }{dT}\right)\mathrm{vs}\;T^{-1}$ for various degree of conversion. Taking into account the use of a linear heating rate (β=dT/dt) and solving the resulting integration, the temperature integral does not provide an exact analytical solution. For this reason, the value of *E_a_* obtained (66.83 ± 2.7 kJ/mol) it is considered to be less accurate than other isoconversional methods differing in their approximations. For example, FWO method, by Doyle’s approximation [37,59] and KAS method through the approximation done by Murray and White leads accurate approximation [60,61]. Finally, the Starink method uses a more accurate approximation of the temperature integral (Figure 4) and, consequently, *E_a_* obtained is 68.57 ± 4.38 kJ/mol with *r* values of −0.992. For all the aforementioned methods, *E_a_* ranges from 66 to 69 kJ/mol, which is in total accordance with most epoxy systems reported by the literature and showing how the use of epoxidized linseed oil as biobased epoxy resin does not affect the conversion (α) of the curing process and a single-step analysis is a valid approximation.

The wide evaluation range (*α* = 0.05–*α* = 0.95) of *E_a_*, provides a high accuracy, although this is only the first approach, since the accuracy of the kinetic analysis depends on other parameters of the kinetic triplet, such as reaction model f(α) and pre-exponential factor, *A*. As summarized in Equations (12) and (13), Malek proposed a method to determine these two parameters by calculating the functions *y*(*α*) and *z*(*α*,) by graphical methods related to the thermally activated process [62]. Figure 5 shows the plot of *y*(*α*) versus the conversion *f*(*α*) at different heating rates. The maximum value of *y*(*α*), denoted as *α_M_*, has been highlighted in the graphic representation. In this case, regardless of the analyzed heating rate, all values obtained are higher than 0 (values listed in Table 4). Due to the values obtained between 0.04 and 0.1 and the geometry of the curves, an autocatalytic reaction model with a high proportion of the autocatalytic process is suggested [63,64]. To fulfil this premise, one of the conditions is that value of *α_M_* is between 0 and *α_p_* (condition clearly fulfilled) and that the maximum of the *z*(*α*) function obtained at different heating rates, denoted as $\alpha_{p}^{\infty }$ is higher than *α_M_*. Figure 6 shows *z*(*α*) plot of experimental data as a function of the degree of conversion by Equation (13), and values of $\alpha_{p}^{\infty }$ are summarized in Table 4.

Once again, regardless of the heating rate used, *α_M_* < $\alpha_{p}^{\infty }$; therefore, the second condition is accomplished to determine that the kinetic process is autocatalytic. Analyzing the evolution of *α_M_* as a function of the heating rate, it is possible to observe a decreasing relationship, with higher values of *α_M_* (0.1038) for a lower heating rate (10 °C/min), indicating that the autocatalytic effect is greater at lower heating rates. This statement agrees with what was found by Lascano et. al. in previous studies [24], where it is stated that the autocatalytic effect is more intense with higher *α_M_* values. Furthermore, as can be seen in Figure 6, the experimental comparison of *z*(*α*) with some generalized master plots [65], shows, according to the particular geometry, a consistent model with a large contribution of the autocatalytic effect.

Thereupon, it is possible to use an autocatalytic reaction model by determining the parameters (*n* and *m*) characteristic of the Šesták–Berggren model (*SB*(*m*,*n*)) as recommended by Malek in the case, accomplished in the actual ELO/MNA system cured at different heating rates, which $\alpha_{p}^{\infty }\neq 0.632$ and $\alpha_{M}\in \left(0,\alpha_{p}\right)$ [40]. This reaction model is governed by Equation (19), when *n* is the *n*th reaction order and the exponent *m* represents the autocatalytic effect, indicating that the conversion rates are directly related to the reacted (crosslinked) material *α*:(19)$$f\left(\alpha \right)=\alpha^{m}\cdot {\left(1-\alpha \right)}^{n}$$

Replacing the function in Equation (4), the following expression is obtained Equation (20):(20)$$\frac{d\alpha }{dt}=A\cdot e^{\frac{-E_{a}}{R\cdot T}}\cdot \;\alpha^{m}\cdot {\left(1-\alpha \right)}^{n}$$

Some authors, like Criado et al. [66] and Malek [40] describe linear iterations to obtain the mentioned *m* and *n* parameters. While other authors like Krishnaswamy et al. [67] and Thaher et al. [68] use non-linear curve fitting to obtain *m* and *n* parameters and the pre-exponential factor, *A* by applying natural logarithms, Equation (21):(21)$$\ln\left(\beta \cdot \frac{d\alpha }{dT}\right)+\frac{E_{a}}{R\cdot T}=\ln\left(A\right)+m\cdot \ln\left(\alpha \right)+n\cdot \ln\left(1-\alpha \right)$$

The values of ln(*A*), *n* and *m* parameters obtained after non-linear curve fitting are summarized in Table 5. Average value of ln(*A*) is 16.924 ± 0.087, and *m* and *n* 0.107 ± 0.065 and 0.973 ± 0.041, respectively, so determination of the kinetic triplet has been carried out. Finally, in order to corroborate the fit of the theoretical model, Figure 7 shows the comparison between the values obtained using the proposed model and the experimental data, represented by symbols. Due to the overlap concordance in both representations, it is possible to state that the kinetic triplet obtained has a good accuracy compared with the experimental data, regardless of the heating rate used.

## 4. Conclusions

The curing kinetics of bio-based epoxy resin from epoxidized linseed oil has been broadly analyzed using dynamic DSC at four different heating rates (10, 20, 30 and 40 °C/min). The apparent activation energy (*Ea*), one of the main kinetic parameters, was determined using differential and integral isoconversional models, giving results with low dispersion [66–69 kJ/mol] and independently of the conversion rate *α*.

The analysis of the highest point in the graphical representation of parameters *y*(*α*) and *z*(*α*) versus conversion, *α*, allowed obtaining the parameters denoted *α_M_* and $\alpha_{p}^{\infty }$, which fulfill two conditions (1: *α_M_* is between 0 and *α_p_* and 2: *α_M_* < $\alpha_{p}^{\infty }$). The higher value of *α_M_* (0.1038), indicates that the autocatalytic effect is greater at low heating rates (10 °C/min). The reaction model, f(α) and the pre-exponential factor, *A*, were obtained through the Šesták–Berggren parameters (*SB*(*m*,*n*)). Theoretical curves plotted applying the values of kinetic triplet obtained show a great concordance with the experimental data, confirming the accuracy of the parameters obtained for an epoxy resin based on ELO. This widens the use of flax crops as linseed oil can be converted into a biobased epoxy resin with similar curing/crosslinking parameters to those of petroleum-derived epoxy resins.

## Figures and Tables

**Figure 1 polymers-13-01279-f001:**
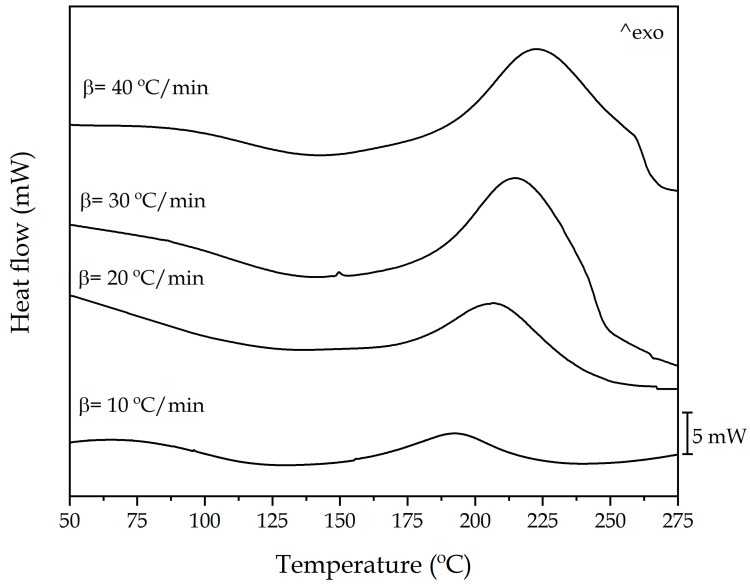
Differential scanning calorimetry (DSC) thermal curves of epoxidized linseed oil (ELO)-methyl nadic anhydride (MNA) system at various heating rates, showing the exothermicity of the crosslinking process.

**Figure 2 polymers-13-01279-f002:**
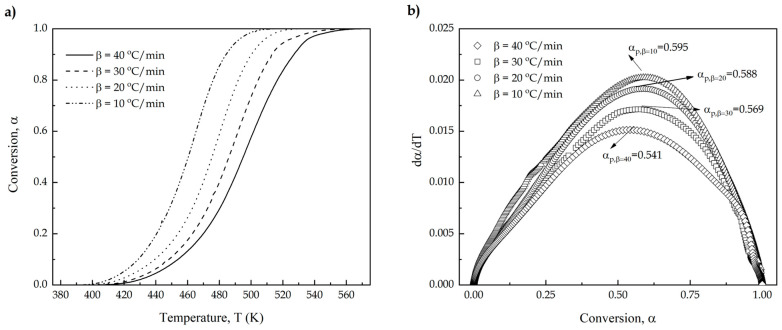
(**a**) the extent of conversion (*α*) and (**b**) the curing rate of an ELO/MNA epoxy system at different heating rates of ELO/MNA epoxy system.

**Figure 3 polymers-13-01279-f003:**
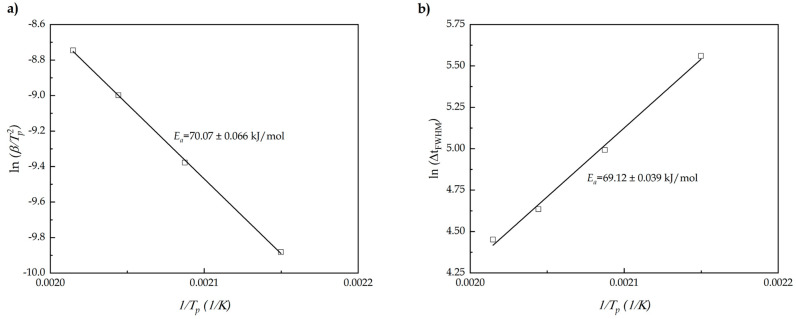
Experimental plot corresponding to (**a**) the Kissinger method and (**b**) Δt_FWHM_ against *1*/(*T_p_*) of an ELO/MNA epoxy system at different heating rates.

**Figure 4 polymers-13-01279-f004:**
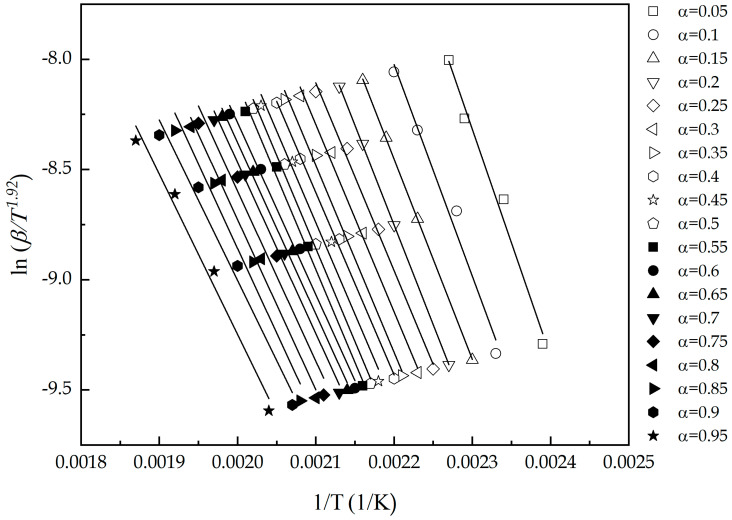
Plot corresponding to Starink method during all the extent of conversion of an ELO/MNA epoxy system at different heating rates.

**Figure 5 polymers-13-01279-f005:**
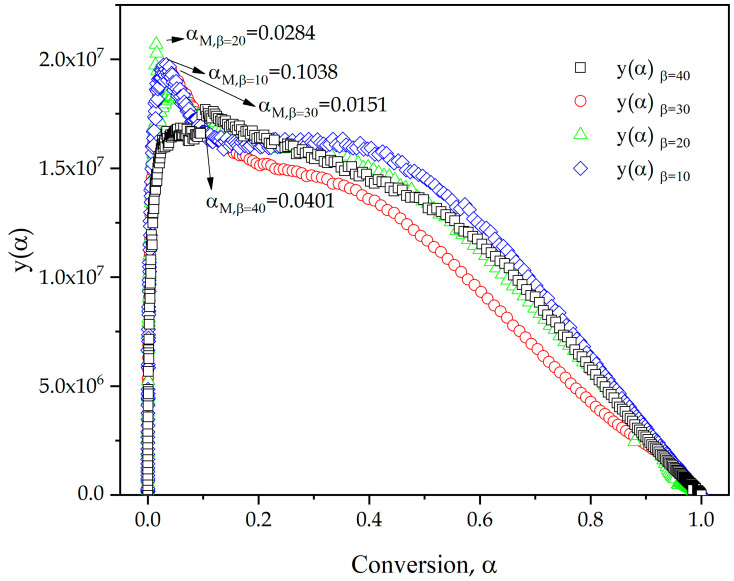
*y*(*α*) plot of experimental data as a function of the extent of conversion of an ELO/MNA epoxy system at different heating rates.

**Figure 6 polymers-13-01279-f006:**
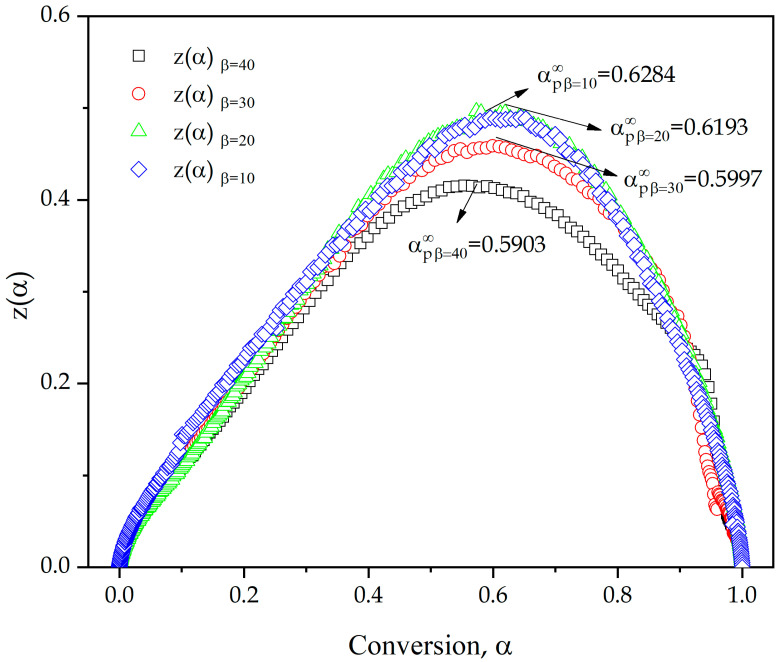
*z*(*α*) plot of experimental data as a function of the extent of conversion of an ELO/MNA epoxy system at different heating rates.

**Figure 7 polymers-13-01279-f007:**
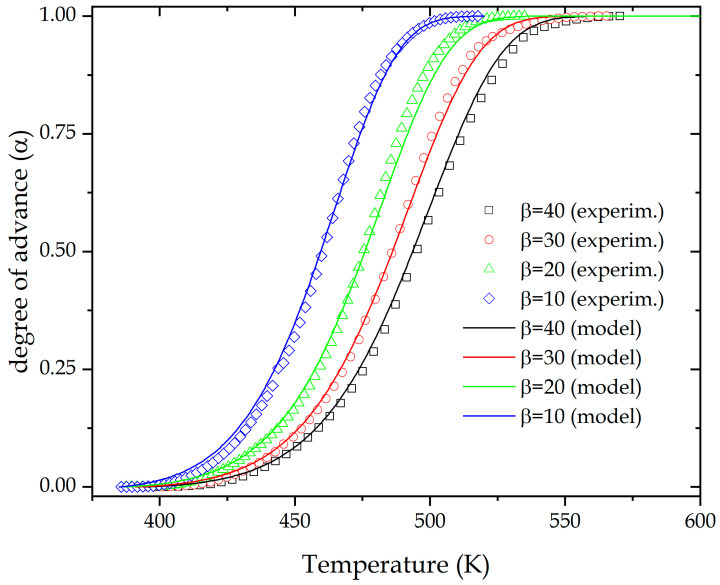
Theoretical and experimental comparison of the extent of conversion of an ELO/MNA epoxy system at different heating rates.

**Table 1 polymers-13-01279-t001:** Summary of algebraic expression for reaction model *f*(*α*).

Reaction Model	Code	*f*(*α*)	
2D-Reaction	R2	(1−α)12	(15)
2D-Diffusion	D2	$\frac{1}{-\ln\left(1-\alpha \right)}$	(16)
Johnson-Mehl-Avrami	JMA(n)	n(1−α)·[−ln(1−α)]1−1n	(17)
Jander	D3	3/2·(1−α)23[1−(1−α)]23	(18)

**Table 2 polymers-13-01279-t002:** Main data obtained from dynamic DSC runs of ELO/MNA epoxy system at different heating rates.

*β* (°C·min^−1^)	Curing Cycle	
	*T_p_* (°C)	Δ*H_T_* (J·g^−1^)
10	192.80	140.72
20	206.70	146.07
30	216.17	198.51
40	223.32	215.77

**Table 3 polymers-13-01279-t003:** Apparent activation energy values obtained by isoconversional methods of an ELO/MNA epoxy system at different heating rates.

Isoconversional Method	Apparent Activation Energy, *E_a_* (kJ/mol)
Friedman	66.83 ± 2.70
Flynn–Wall–Ozawa (FWO)	66.27 ± 3.56
Kissinger–Akahira–Sunose (KAS)	66.22 ± 2.62
Starink	68.57 ± 4.38

**Table 4 polymers-13-01279-t004:** Maximum values obtained from *d*(*α*)/*dT*, *y*(*α*) and *z*(*α*) function of an ELO/MNA epoxy system at different heating rates.

*β* (°C/min)	$\alpha_{p}$	$\alpha_{M}$	$\alpha_{p}^{\infty }$
10	0.595	0.1038	0.6284
20	0.588	0.0284	0.6193
30	0.569	0.0151	0.5997
40	0.541	0.0401	0.5903

**Table 5 polymers-13-01279-t005:** Calculated *SB*(*m*,*n*) parameters by nonlinear fit of a ELO/MNA epoxy system at different heating rates.

*β* (°C/min)	*ln*(*A*)	*m*	*n*
10	17.015 ± 0.011	0.185 ± 0.041	1.011 ± 0.007
20	16.939 ± 0.018	0.127 ± 0.006	0.916 ± 0.011
30	16.939 ± 0.019	0.0283 ± 0.006	0.975 ± 0.013
40	16.805 ± 0.019	0.089 ± 0.006	0.993 ± 0.013
Average Value	16.924 ± 0.087	0.107 ± 0.065	0.973 ± 0.041

## Data Availability

The data presented in this study are available on request from the corresponding author.

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
