# Peer review of "Kinetic Analysis of the Curing Process of Biobased Epoxy Resin from Epoxidized Linseed Oil by Dynamic Differential Scanning Calorimetry"

_polymers, 2021, doi:10.3390/polym13081279_

Round 1

Reviewer 1 Report

The paper is well written and deserves publication. Just for better reading, I’d suggest following trivial things;

Page-1: Line-3: ‘isocoversional’ should be ‘isoconversional’.

Page-1: Line-4: Not sure if double commas are required. If yes, please ignore.

Page-1: Line-2: ‘consistence’ should be ‘consistency’.

Page-1: Line-9: ‘clothing manufacturing’ seems fine but I’d prefer ‘clothes manufacturing’.

Page-1: Line-5: ‘fiberglass’ should be ‘fiber glass’.

Page-2: Line-1: ‘modification’ should be ‘modifications’.

Page-3: Line-3: ‘was of set to’ should be ‘was set to’.

Page-3: Line-4: Not sure if parenthesis are empty by intention.

Page-3: Line-8: Not sure if (40   L) unit is correct. Would be a too large crucible!

Page-4: Line-6: Delete ‘they’.

Page-4: Line-8: ‘no’ should be ‘not’.

Page-7: Line-3: I believe ‘It’ is not required.

Page-7: Line-1: Not sure if parenthesis are empty by intention.

Page-8: Line-1: ‘use isoconversional’ should be ‘use of isoconversional’.

Author Response

In the attached word document, reviewer can find a response point-by-point to te reviewer´s comments. 

Reviewer 2 Report

In this work, the authors have described the curing reaction of epoxidized linseed oil with anhydride and determined the kinetic parameters using DSC. There are a few questions/suggestions which need to be addressed.

The most important concern is that this paper does not bring out the novelty of the work. There are many papers published on the kinetic studies of epoxies by DSC already, and epoxy-anhydride systems are also well studied.

1. Introduction section -

    1. Three paragraphs on flax are not required. A few lines, mentioning how it is connected to linseed oil, are sufficient.
    2. The authors should provide more background on their work and highlight the importance of the work. Lots of research have been published on the kinetic analysis of epoxy curing. How is this work different from the others? What other papers have been done on the curing of ELO?

2. Materials section- Why was anhydride chosen as the hardener? Line 119- There are many other hardeners that are liquid, and they form a homogeneous mixture with epoxies like amines, phenols. Why was anhydride used?

3.Line 128 - 40 L cannot be the volume of the crucible.

4. The degree signs are underlined.

5. Lines 239-249- Please explain how your results compare to the literature.

6. Line 246 - Please explain the overlapping processes.

Author Response

(The authors gave the same response as above.)

Round 2

Reviewer 2 Report 

The authors have answered my queries satisfactorily.

Author Response

Thank you very much for your comments. In the attached document, the authors would like to thank the reviewer for considering the article suitable for publication. 
